# High Efficiency Mercury Sorption by Dead Biomass of *Lysinibacillus sphaericus*—New Insights into the Treatment of Contaminated Water

**DOI:** 10.3390/ma12081296

**Published:** 2019-04-19

**Authors:** J. David Vega-Páez, Ricardo E. Rivas, Jenny Dussán-Garzón

**Affiliations:** 1Microbiological Research Center (CIMIC), Department of Biological Sciences, Universidad de Los Andes, Bogotá 111711, Colombia; jd.vega1754@uniandes.edu.co; 2Department of Chemistry, Universidad de Los Andes, Bogotá 111711, Colombia; re.rivas@uniandes.edu.co

**Keywords:** mercury, biosorption, dead cells, *Lysinibacillus sphaericus*, dithizone, GF-AAS, EDS-SEM

## Abstract

Mercury (Hg) is a toxic metal frequently used in illegal and artisanal extraction of gold and silver which makes it a cause of environmental poisoning. Since biosorption of other heavy metals has been reported for several *Lysinibacillus sphaericus* strains, this study investigates Hg removal. Three *L. sphaericus* strains previously reported as metal tolerant (CBAM5, Ot4b31, and III(3)7) were assessed with mercury chloride (HgCl_2_). Bacteria were characterized by scanning electron microscopy coupled with energy dispersive spectroscopy (EDS-SEM). Sorption was evaluated in live and dead bacterial biomass by free and immobilized cells assays. Hg quantification was achieved through spectrophotometry at 508 nm by reaction of Hg supernatants with dithizone prepared in Triton X-114 and by graphite furnace atomic absorption spectroscopy (GF-AAS). Bacteria grew up to 60 ppm of HgCl_2_. Non-immobilized dead cell mixture of strains III(3)7 and Ot4b31 showed a maximum sorption efficiency of 28.4 µg Hg/mg bacteria during the first 5 min of contact with HgCl_2_, removing over 95% of Hg. This process was escalated in a semi-batch bubbling fluidized bed reactor (BFB) using rice husk as the immobilization matrix leading to a similar level of efficiency. EDS-SEM analysis showed that all strains can adsorb Hg as particles of nanometric scale that can be related to the presence of S-layer metal binding proteins as shown in previous studies. These results suggest that *L. sphaericus* could be used as a novel biological method of mercury removal from polluted wastewater.

## 1. Introduction

Mercury (Hg) is a highly toxic metal widely dispersed and because of its frequent use in several industries, it is a serious cause of poisoning due to bioaccumulation [1]. Symptoms of Hg poisoning are usually nonspecific and include fatigue, anxiety, depression, paresthesia, weight loss, memory loss, difficulty concentrating, and fetus malformation [2,3,4], but most importantly, these are usually manifested after months or years of continuous exposure to low doses of the metal [4,5].

The cycles followed by Hg are linked to liquid sources [6,7] and, according to their interaction with other compounds and organisms, they can change their oxidation state [6,8,9]. However, Hg methylation is one of the most important processes, since organic Hg compounds are the most toxic for humans [10,11,12]. Although methylmercury slowly undergoes demethylation processes [13] it accumulates in the brain, liver, kidneys, placenta and brain of the fetus, peripheral nerves, and bone marrow [14] causing damage.

In 2013, most countries worldwide signed the Minamata Convention on Mercury [15]. However, several places have reported Hg contamination events through these years [16,17,18,19] and Colombia’s high rates in artisanal gold mining using Hg [20,21,22] are proportionally related to reports of clinical cases [23,24]. In fact, in 2012, it was reported that the population of Segovia in Antioquia, Colombia, had the highest per capita Hg contamination in the world, due to artisanal gold extraction [22] and other cases of people with high levels of Hg have been widely documented [25,26].

There are various physicochemical methods to treat water contaminated with Hg and research is currently focused on the development of nanomaterials to detect and remove mercury [27,28,29]. Different types of reactors are often escalated and coupled to classical techniques such as filtration, chemical coagulation, or sedimentation which can remove up to 80% of inorganic Hg and 40% of organic Hg [30], but all those techniques are often considered expensive because of the materials used in the design of the membranes or the reactants used in metal complexation. Despite problems in the escalation process [31], bubbling fluidizing bed reactors (BFB) are often a good alternative in biological treatments of high volumes of water samples contaminated with metals [32,33].

Biological treatment of organic Hg has been tested mainly using bacteria with alkylmercury lyase (MerB) which can transform organic Hg into the inorganic form [34]. The structure of this enzyme has been extensively studied as has its kinetics [35]. However, most of these microorganisms present risks in pathogenicity for humans—such as *Staphylococcus aureus*, *Shigella flexneri*, *Escherichia coli*, and *Pseudomonas sp*. [36]—so their use is not recommended in bioremediation.

Microorganisms have several types of interactions with Hg due to specific biochemical characteristics of the species such as the availability of sulfur-rich functional groups on their surface, or specific proteins involved in transport and/or oxidation-reduction processes of Hg species [34,35,36,37,38]. These types of interactions are usually classified as tolerance and resistance, where “tolerant” interactions are those in which the microorganism is able to capture (adsorb) the metal but not to metabolize it, and the “resistant” interactions can transform the metal from one species to another, often by redox reactions at intracellular level after being absorbed [39,40].

Tolerance of some bacilli strains to Hg was reported [41,42] and later it was identified that *Lysinibacillus sphaericus*—an aerobic gram positive spore-forming bacterium, nonpathogenic in humans—is able to adsorb metals [43] such as iron (Fe), cadmium (Cd), arsenic (As) [44], chromium (Cr), and lead (Pb) [45]. This demonstrates its tolerance and resistance to toxic metals by a mechanism which is known as crossed regulation [46]. Also, it was recently proven that *L. sphaericus* can adsorb gold (Au) and probably even synthesize nanoparticles [47].

Furthermore, dead cells of different *L. sphaericus* strains have also been proven as efficient metal accumulators by passive process [48] and the presence of S-layer proteins in metals binding has been widely discussed for the genus [49,50,51] even in sporulated forms. The advantages of dead over live cells in biological treatment of pollutants include a lack of risk to the environment and no need to be preserved since binding mechanisms are non-metabolic [52,53,54]. Thus, the objective of this work was to establish the Hg removal capability of three different dead *L. sphaericus* strains to assess its possible use in the treatment of water contaminated with this metal.

## 2. Materials and Methods

The three Colombian *L. sphaericus* strains from the CIMIC culture collection used in this study are shown in Table 1. All the strains were grown in nutrient agar for 24 h at 30 °C. A colony of each strain was incubated in 20 mL of nutrient broth for 24 h at 30 °C before the assessment of Hg minimum inhibitory concentration (MIC).

### 2.1. Mercury Removal by Free Cells

A selective pressure with Hg was performed to identify minimum inhibitory concentration (MIC) on all *L. sphaericus* strains using nutrient broth (NB) and minimal salt medium supplemented with sodium acetate and yeast extract (MSM). Bacteria growth was evaluated in HgCl_2_ solutions from 5 mg/mL to 80 mg/L. Growth was determined with an OD600 of at least 0.1 in less than 10 days.

Live and dead *L. sphaericus* cells were tested to differentiate between absorption and adsorption mechanisms. Strains were used alone and mixed. 100 µL of each strain growth at the highest HgCl_2_ concentration were taken and grown in an overnight culture (ON) of 100 mL NB at 30 °C under agitation at 150 rpm to late exponential growth (OD600 between 0.5 and 0.8 monitored by a UV–vis BioMate 3 spectrometer, Thermo Scientific, Roskilde, Denmark). Cells were then centrifuged at 11,500 rpm for 30 min at 4 °C and pellets were washed three times with deionized water. Dead bacteria were obtained by treating pellets with 10% formalin in phosphate-buffered saline (PBS) for an hour [59] and dead pellets samples were cultured in nutrient agar to verify unviability.

Live and dead pellets were dissolved in 40 mL of HgCl_2_. Bioassays were incubated under agitation at 30 °C and 150 rpm. 10 mL aliquots were taken at 5, 10, 15 and 60 min and centrifuged for 2 min. Supernatants were also filtered through 0.22 µm pore and quantified through UV–vis spectrophotometry. Removal assays were made in triplicate with two biological replicates.

### 2.2. Mercury Removal by Immobilized Dead Cells

The most efficient treatment found among non-immobilized cells was used in an escalation process. Dead cells were immobilized in sterile rice husk (RH) and a filter filled with RH was designed to be used in a semi-batch bubbling fluidized bed reactor (BFB) (Appendix A). BFB was supported by a secure vessel in case of foam excess production during bubbling process. The set point was established as Hg concentration: 60 ppm. Operative volume was 8 L of HgCl_2_ loaded at time 0 working at 30 °C. Treatment was performed over 5 days with a residence time of 24 h and 50% recycle. Aliquots were taken each day and quantified through GF-AAS.

### 2.3. Mercury Quantification in Free Cells

Mercury quantification in supernatants of non-immobilized cell treatments was performed by producing a mercury dithizonate complex in Triton X-114 (Hg-Dz), which could be read by absorbance measures using a UV–vis spectrometer (BioMate 3S Spectrophotometer, Thermo Fisher Scientific, Waltham, MA, USA). Preliminarily, dithizone solutions (Dz) were prepared by dissolving 5 mg in different solvents (methanol, ethanol, butanol, toluene, carbon tetrachloride, chloroform, dichloromethane, and micellar medium of Triton X-114) to identify optimum polarity medium whose absorption spectrum of dithizone would not interfere with Hg-Dz complex at the supernatant pH. Reaction was read at 508 nm after 30 s. This colorimetric method was validated by inductively coupled plasma optical emission spectrometry (ICP-OES, iCAP 6500, Thermo Scientific, Waltham, MA, USA) under routine conditions for mercury analysis. All working standard solutions were prepared using Hg Panreac standard (lote: R3861502) and measurements were made in triplicate.

### 2.4. Mercury Quantification in Immobilized Cells

Aliquots from the escalated process were measured by atomic absorption spectrometry with electrothermal atomization because yellow color in the samples after treatment with RH could generate interference in the Hg-Dz complex spectrum. High-resolution continuum source atomic absorption spectrometer (RH-CSAAS, ContrAA 800, commercially available from Analytik Jena, Jena, Germany) in graphite furnace mode was used (GF-AAS). To avoid mercury volatilization during heating steps, 1% palladium (m/v) (Merck) dilution was used as a modifier.

### 2.5. EDS-SEM Analysis

10 µL of diluted bacteria samples from the free cells assays were taken and allowed to dry on a coin covered with sterile aluminum foil. This was then analyzed at the Uniandes Microscopy Center by SEM observation and metal semi quantification with EDS using a Tescan LYRA 3 Scanning electron microscope (TESCAN, Brno, Czech Republic). Qualitative results were compared with a control: *Escherichia coli* K12 C600.

RH (Rice Husk) was also observed by drying samples at 30 °C for 6 h and applying a gold sputter coating to inhibit charging, reduce thermal damage, and improve the secondary electron signal required for topographic examination of bacteria and HR in SEM.

### 2.6. Statistical Analysis

Minitab^®^ v.17 software was used for the statistical analysis of data. Analysis of Variances (ANOVA) was used to determine whether there were significant statistical differences between treatments. Analysis of Means (ANOM) graphs were obtained to show the effects of variables (strains: III(3)7, Ot4b31, CBAM5, mixtures of two or three strains; and state: live or dead) in mercury removal among treatments.

## 3. Results

### 3.1. Minimum Inhibitory Growth Concentration (MIC)

Minimum inhibitory concentration is defined as the minimum concentration of an antimicrobial agent, to which a microorganism does not show evident growth [60]. Table 2 shows the results of the selection pressure and highlights the MIC for the strains evaluated.

*L. sphaericus* highest growth concentration with HgCl_2_ was at 60 mg/L achieved in NB for all strains. These bacteria were stored as metal tolerant strains and used in further assays.

### 3.2. Mercury Removal by Non-Immobilized L. sphaericus Cells

Hg spectrophotometric quantification through Dz was made and the detection limit of the method was in the range of 2 ppm to 4 ppm for calibration curves between 2 ppm and 10 ppm, and bacteria wet weight was determined to calculate Hg removal efficiency (Figure 1 and Appendix A). Spectrophotometric quantification showed no significant differences with ICP-OES (Figure 2).

Dz reaction with Hg forms an orange color complex which can be read by spectrophotometry in the range of 488–510 nm [61]. Excess dithizone was necessary to acquire a broad range of linearity in calibration curves for Hg determination by achieving a quantitative reaction throughout the linear working range (Appendix A). Dz color must be emerald green in order to get absorption peaks that do not interfere with Hg-Dz. A pH of 5–6 was optimal for Hg-Dz formation in excess of dithizone since, at that pH, there is no background interference (Appendix A).

Dead bacteria showed higher Hg removal efficiency than live bacteria in all treatments except for strain III(3)7. Mixtures of two strains showed higher efficiency results compared to each strain. Dead *L. sphaericus* III(3)7 + Ot4b31 cells showed a maximum removal of 32.85 µg/mg bacteria after an hour and 28.4 µg/mg bacteria in less than 5 min (more than 95% of Hg). A mixture of three strains showed lower efficiency behavior than the rest of treatments (Figure 3).

Efficiency differences in Hg removal among strains, can be associated with S-layer protein quantity in these bacteria and by synergic effects between living cells as will be discussed later.

### 3.3. Mercury Removal by Immobilized Mixture of Dead L. sphaericus Strains III(3)7 and Ot4b31 on RH

RH with and without bacteria was used in a BFB reactor to test Hg removal in a proportion of 5-times more Hg than bacteria. AAS has a high-resolution monochromator which identifies possible spectral interferences caused by the sample matrix at the wavelength of the most sensitive working atomic line for Hg, 253.652 nm. Signal detection showed no background interference (Figure 3: top left).

Mercury removal efficiency with RH and bacteria was 2.64 times higher than RH alone (control) as shown in Figure 3 and Table 3. However, the effect of the RH matrix was eliminated so the net effect of dead mixture of *L. sphaericus* III(3)7 and Ot4b31 was calculated as being 18% less efficient than the non-escalated process (Table 3).

### 3.4. EDS-SEM

The analysis of energy dispersion spectroscopy coupled with the scanning electron microscopy (EDS-SEM), shows the cells on a micrometric scale, detecting the most abundant elements in the generated image. Metals can be easily detected by high-intensity reflecting electrons irradiated on the sample.

EDS-SEM shows that mercury is accumulated as spherical particles on *L. sphaericus* (Figure 4 and Appendix A) while this attachment behavior is not observed on surface of *E. coli* (Appendix A). There is also a possibility that Hg could be absorbed by bacteria, but a dense peptidoglycan cell wall prevents the beam from being reflected. Dot pattern found in *L. sphaericus* III(3)7 after an hour of contact with Hg was also observed in strains Ot4b31 and CBAM5 and no differences between living and dead cells were observed (Figure 5).

RH was also observed after Hg treatment to verify cell attachment to RH surface and Hg sorption in cells. Hg dot pattern found in non-immobilized cells was also found on the immobilized ones. RH alone showed large Hg particles attached to its surface (Appendix A).

### 3.5. Statistical Analysis

ANOM analysis showed a confidence interval type of approach that allowed us to determine which, if any, of the levels had a significantly different mean from the overall average of all the group means combined. Figure 6 shows that both the strain and the treatment factors are significant in this study since mercury removal is clearly improved using dead cells and mixtures of *L. sphaericus* III(3)7 + Ot4b31 or III(3)7 + CBAM5 strains.

## 4. Discussion

Colombia is a country where illegal and artisanal goldmining has led to high rates of Hg contamination throughout over 60% of the country [62]. Even though Colombia signed the Minamata Convention on Mercury in 2013, it was only last year that the government set the law to reduce Hg reduction over the next 5 years [63]. However, Hg pollution continues to be a problem, especially for mining communities where water treatment is difficult to achieve due to their political and economic contexts. In fact, no treatment is currently implemented in any of the affected communities.

Biological treatment of Hg has been investigated using microorganisms that are easily found in polluted sediments, but sorption mechanisms must be further studied in order to establish whether Hg particles are transformed into more toxic or difficult to handle species or not. Hg^0^ can be rapidly volatilized by reductases codified by merA genes or sulfate-reducing mechanisms [64,65]. Methylated species are the most dangerous species and biological systems tends to methylate Hg when hgcAB genes are expressed [66]. Therefore, Hg removal by tolerant microorganisms capable of adsorbing the heavy metal is highly desirable but transforming it by resistance mechanisms is not.

Biosorption can be defined as the property of biological materials to accumulate heavy metals from aqueous polluted solutions through physicochemical pathways of uptake or by binding and concentration of heavy metals from even very dilute amounts [67,68]. Since active transport of metals requires energy and is not performed by dead biomass, bioaccumulation by absorption mechanisms, which leads to potential changes in mercury species characteristic from resistant bacteria, cannot happen.

HgCl_2_ solution is mainly composed by irregular Hg salt crystals, so spheres morphology is achieved by an unknown mechanism on the surface of *L. sphaericus* cells. Particle sizes ranged between 70 nm and 120 nm so possible formation of nanoparticles could be occurring if there was an active Hg reductase protein being expressed in the cells. However, no merA genes have been annotated in the genomes of all three strains, nor have merB or hgcAB genes that codify to alkyl mercury lyase and mercury methylase, respectively [43].

Gómez and Dussan [43] proposed a metabolic model overview based on annotated *L. sphaericus* toxic lineage genomes where there are some specific transport proteins for metals such as As/Sb, Fe, and Cr among others. However, Hg complexation with sulfhydryl rich molecules—such as cysteine from proteins—has been shown to facilitate Hg entering the cells [69], and that could be a reason for the size of the Hg particles found in EDS-SEM results.

However, chemical interactions of metals in biological models are often unspecific since most of them can be transported by different channels. MerT transporter has not been identified in *L. sphaericus* strains so there could be a lack of *mer* operon that would lead to Hg specific transformation. Hg reduction could be occurring also by presence of quinones and/or other molecules that could react spontaneously with Hg [70,71]. That would also explain the size of Hg particles found both in living and dead cells in EDS-SEM images.

Results of Hg removal by living and dead bacteria are consistent with previous studies with other metals where it is a fact that dead bacteria tend to be more efficient than living bacteria [42,52,72]. This could be due to active transport mechanisms on living bacteria that constantly uptake and desorb Hg by unspecific channels. Moreover, it is necessary to highlight that Hg is not accumulated in cytoplasm since resistance genes from the *mer* operon are absent.

Higher efficiency of mixed strains both in live and dead treatments is probably related to the amount of S-layer proteins in each strain since III(3)7 has 13 full copies, Ot4b31 has 32 copies and CBAM5 has 21 copies but most of them are truncated [56,57,58]. These S-layer proteins are usually immersed in an exopolysaccharide matrix as shown by Francois and collaborators [72] that could be making high complexity structures by molecular interactions.

Also, it is important to notice that CBAM5 is the only strain isolated from a heavily polluted environment, which may have specific genes to degrade carbohydrates instead of specific metals tolerance like Hg. This is unusual in petroleum exploitation areas [73] and all evolutive pressures of this strain could be a reason for low affinity when working with other strains, even with unspecific interactions. The negative synergy of this strain in Hg removal requires further research.

Dead bacteria present several advantages because of their lack of metabolic susceptibility to changes from the environment by undesired horizontal gene transfer from other microorganisms [74], and also because they can be more easily accepted for people than “live” bacteria since all microorganisms tend to be more related to pathogens and illness [75], especially among Hg affected rural communities due to their low access to education [76]. Dead bacteria have proven to be efficient in mercury removal, but Colombian strains tested in this study against Hg concentrations as high as 60 ppm showed a promising inexpensive method to provide water treatment in communities that are still receiving mercury pollution from mining activities.

A pilot BFB reactor proved that an agricultural residue like RH is a good matrix to immobilize dead bacteria and to improve Hg removal efficiency from polluted water. This matrix could later be easily disposed of since solid wastes are easier to handle and treated water could be less harmful to communities. Further studies are needed to implement an in-field removal process by adjusting parameters to ensure a maximum concentration of 1 ppb in treated water.

## 5. Conclusions

*L. sphaericus* is a well-known bacterium capable of metal adsorption and absorption. Colombian strains III(3)7, Ot4b31, and CBAM5 have been widely tested with metals such Cd, Fe, Pb, As, Co, Cr, Zn and, more recently, with Au. However, this is the first Hg wide study using these strains with an important result in Colombia’s context of water pollution due to illegal and artisanal mining. Both dead and live cells can be used in Hg removal but dead cells of mixed strains III(3)7 and Ot4b31 are the most efficient.

Also, immobilization in RH matrix—often an agricultural residue—provided a good yield of removal in a BFB pilot reactor. These results support the propose of a novel clean method to purify heavily polluted water at low cost as an alternative to expensive traditional methods. Moreover, this could be the first approach to solving the current issue of Hg pollution in Colombia’s rivers. Studies conducted in the field or with water samples from affected zones are necessary to assess bacterial behavior in non-synthetic mercury polluted samples since more elements and molecules could improve or interfere in the high efficiency of the proposed method.

## Figures and Tables

**Figure 1 materials-12-01296-f001:**
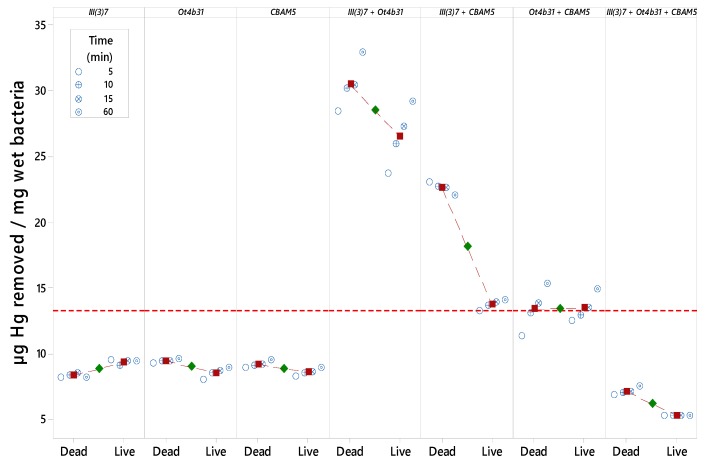
Efficiency in mercury removal by *L. sphaericus* (strains alone and mixtures) over time.

**Figure 2 materials-12-01296-f002:**
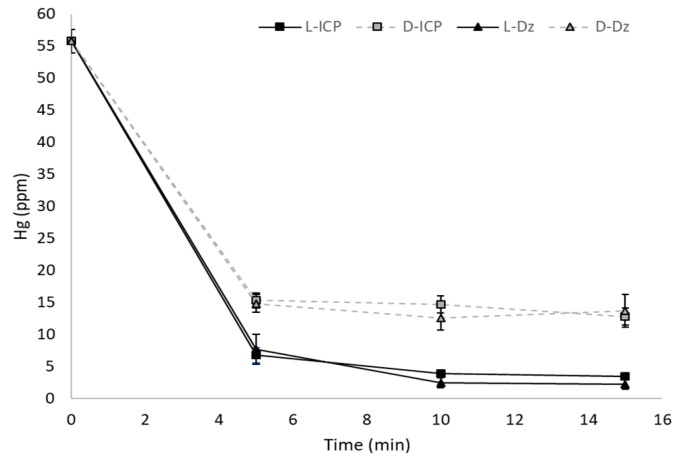
Comparison of Hg quantification with ICP-OES. Data of *L. sphaericus* III(3)7 treatment. L: Live; D: Dead.

**Figure 3 materials-12-01296-f003:**
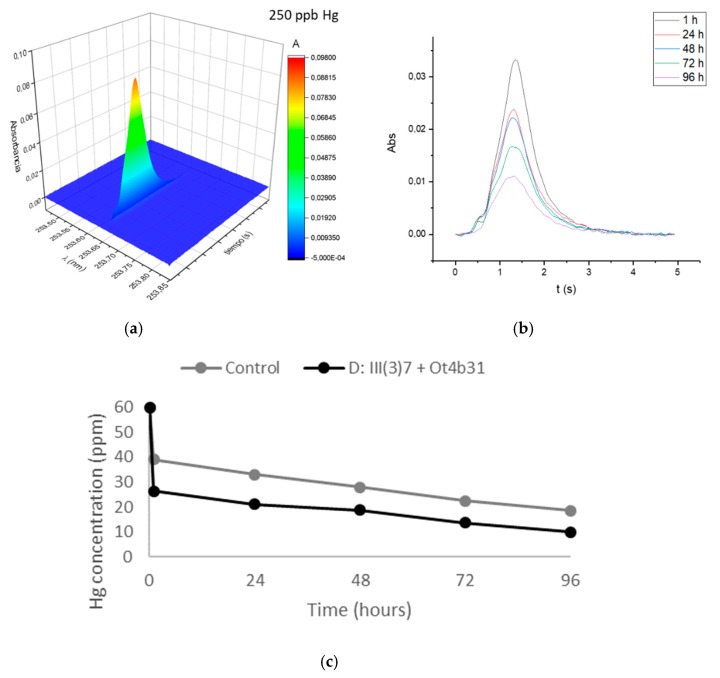
GF-AAS quantification of Hg aliquots of escalated process. (**a**) Hg signal of standard solution diluted 1/240. (**b**) Time-resolved absorbance signals for samples. (**c**) Hg concentration over time in escalated process with (black) and without bacteria (grey).

**Figure 4 materials-12-01296-f004:**
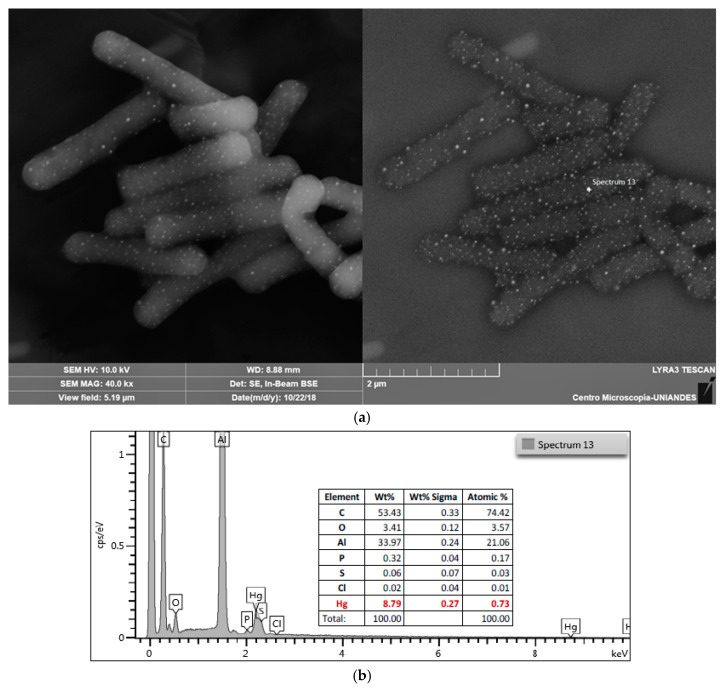
EDS-SEM analysis of *L. sphaericus* III(3)7 live cells after 1 h in contact with HgCl_2_. (**a**) Image taken at 10 kV by detection of secondary electrons and backscattered electrons; (**b**) Data from energy dispersive spectroscopy of punctual area “Spectrum 13”.

**Figure 5 materials-12-01296-f005:**
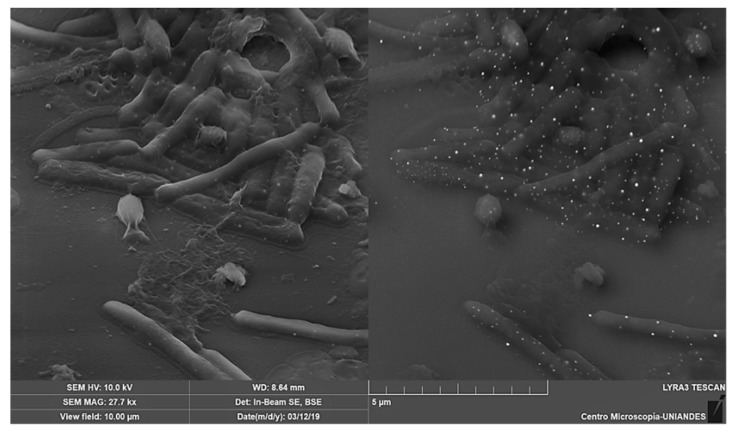
EDS-SEM of dead *L. sphaericus* III(3)7 and Ot4b31 immobilized in RH after in contact with HgCl_2_.

**Figure 6 materials-12-01296-f006:**
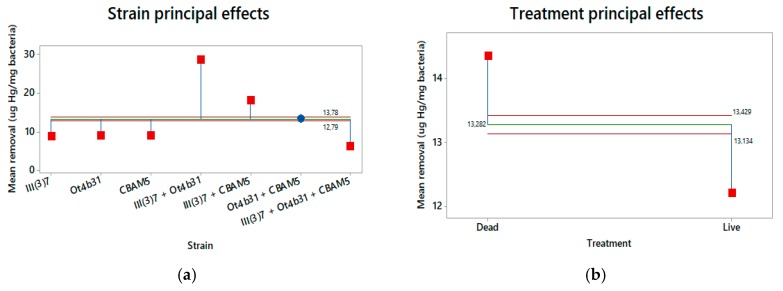
ANOM graphs of principal effects on mercury removal by *L. sphaericus.* (**a**) Effect of bacteria strain (alone or mixed); (**b**) Effect of bacteria treatment (dead or live).

**Table 1 materials-12-01296-t001:** *Lysinibacillus sphaericus* strains used.

Strain	Origin	Ref.Seq Assembly Accession Number	Metals Removal References
III(3)7	Soil sample from oak forest	GCF_001598075.1	[44,46,55,56]
Ot4b31	Larvae, Beetles	GCF_000392615.1	[47,48,57]
CBAM5	Subsurface soil sample of petroleum exploration area	GCF_000568835.1	[47,55,58]

**Table 2 materials-12-01296-t002:** MIC of HgCl_2_ on *L. sphaericus* strains.

*L. sphaericus* Strain	Liquid Culture Media (MSM/NB)					
	5 ppm	10 ppm	20 ppm	40 ppm	60 ppm	80 ppm
III(3)7	+/+	+/+	−/+	−/+	−/+	−/−
Ot4b31	+/+	−/+	−/+	−/+	−/+	−/−
CBAM5	+/+	+/+	−/+	−/+	−/+	−/−

− No Growth; + Growth

**Table 3 materials-12-01296-t003:** Mercury removal efficiency comparison of free cells (after 1 h) vs. escalated process.

Treatment	Mercury Removal Efficiency (µg Hg/mg bacteria)
Non-escalated mixture	32.85
Mixture with RH	71.76
Mixture without RH effect (normalized)	27.04

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
