# Peer review of "High Efficiency Mercury Sorption by Dead Biomass of Lysinibacillus sphaericus—New Insights into the Treatment of Contaminated Water"

_materials, 2019, doi:10.3390/ma12081296_

Round 1

Reviewer 1 Report

Generally, the manuscript was prepared correctly. Methodology and analysis of results rather don't raise any objections. However, the obtained results do not contribute significantly to the existing state of knowledge. The paper remains rather a listing of results than an integration of the information. An assessment putting the findings into perspective and make a solid conclusion is missing. The authors should emphasize more the novelty and usefulness of the results. Moreover, discussion should be extended. There are many papers on this topic that the authors should take into consideration when they discuss the obtained results.

Author Response

Thanks for your comments.

In the attached Word document we response to your appreciations. There you will find a cover letter explaining changes done and next you will find the manuscript with highlighted modifications.

Reviewer 2 Report

Comments on the manuscript entitled “High Efficiency Mercury Sorption by Dead Biomass 2 of Lysinibacillus Sphaericus. New Insights into the 3 Treatment of Contaminated Water”.

The manuscript presents the elimination of mercury using live and dead bacterial biomass. The bacteria reported are adequately characterized, and in general, the conclusions are adequately supported by the data presented. Only minor modification might further improve the manuscript. The following point should be addressed:

On page 3 line 128, continuous should be spelt continuum.

Author Response

(The authors gave the same response as above.)
